# Comprehensive Oral Diagnosis and Management for Women with Turner Syndrome

**DOI:** 10.3390/diagnostics14070769

**Published:** 2024-04-05

**Authors:** Victoria Tallón-Walton, Meritxell Sánchez-Molins, Wenwen Hu, Neus Martínez-Abadías, Aroa Casado, María Cristina Manzanares-Céspedes

**Affiliations:** 1Human Anatomy and Embryology Unit, Experimental Pathology and Therapeutics Department, University of Barcelona, 08907 Barcelona, Spain; vtallon@ub.edu (V.T.-W.); wenwenhu.02@gmail.com (W.H.); 2Odonto-Stomatology Department, University of Barcelona, 08907 Barcelona, Spain; meritxellsanchez@ub.edu; 3Evolutionary Biology, Ecology and Environmental Sciences Department, University of Barcelona, 08007 Barcelona, Spain; neusmartinez@ub.edu

**Keywords:** Turner syndrome, oral health, craniofacial phenotype, tooth anomalies, anatomy

## Abstract

Turner Syndrome (TS) is a rare genetic disorder that affects females when one of the X chromosomes is partially or completely missing. Due to high genetic and phenotypic variability, TS diagnosis is challenging and is often delayed until adolescence, resulting in poor clinical management. Numerous oral, dental and craniofacial anomalies have been associated with TS, yet a comprehensive description is still lacking. This study addresses this gap through a detailed analysis of oral health and craniofacial characteristics in a cohort of 15 females with TS and their first-degree relatives. Subjects with TS ranged from 3 to 48 years old, none showed evidence of periodontal disease and only the youngest was in mixed dentition. Using the Multifunction System, we identified an aggregation of multiple signs and symptoms in each TS subject, including tooth anomalies (supernumerary molars, agenesis, microdontia, enamel defects, alterations in eruption patterns -advanced and delayed for chronological age-, crowding, rotations and transpositions), malocclusion (class II/1 and II/2) and Class II facial profile, while relatives exhibited fewer manifestations. The early detection of these signs and symptoms is crucial for appropriate referral and the optimal clinical management of TS, especially during the critical period of 9 to 10 years when congenital dental anomalies appear. The use of an established taxonomy to describe these phenotypic features is essential for early detection. Multidisciplinary teams are required to ensure holistic care management in rare diseases like TS.

## 1. Introduction

Turner Syndrome (TS) is characterized by either a total or partial absence of the second sex chromosome (X chromosome) or by a structural defect in one of the X chromosomes in female subjects [1,2,3]. Due to its low prevalence, being found in approximately 1 in every 2000/2500 live-born girls, TS is considered a rare disease. TS is also reported as a common cause of miscarriage in the early stages of pregnancy [1,3,4]. Although Turner syndrome is among the most common syndromes associated with sex chromosomes and is well recognized within the health practitioner community, TS diagnosis remains challenging and is often delayed until adolescence. In Spain, the mean average age of diagnosis is 15 years old. In numerous instances, Turner syndrome goes undiagnosed until women encounter fertility issues, while some cases remain entirely undetected [3,5,6,7]. Due to misdiagnosis or delayed diagnosis, many TS women lack proper hormonal treatment and healthcare management, which leads to a diminished quality of life.

The phenotypic characteristics of TS include primary hypogonadism, skeletal anomalies, congenital heart diseases, gastrointestinal and hepatic pathologies, visual and auditory problems, lymphatic and cutaneous ailments, renal malformations and hypertension, as well as mental, neurocognitive and behavioral health issues [8,9]. However, the phenotype is highly variable and the precise etiology of each trait associated with TS remains unclear, probably because the TS karyotype greatly differs between subjects [8]. Currently, there is a limited understanding of the genotype–phenotype correlation in TS [10], underscoring the need for further research to identify additional phenotypic factors and genetic mechanisms that explain the heterogeneity of TS signs and symptoms [9].

The complexity and heterogeneity of the phenotype requires a highly specialized multidisciplinary team to adequately address the clinical management of women with TS, with a holistic approach that tackles all the clinical and non-clinical aspects of TS [11,12]. The current management of TS includes hormone replacement therapy (HRT), growth hormone (GH) treatment and the treatment of associated comorbidities [8,11,12].

In recent years, there has been an increasing interest in the oral health and craniofacial phenotype associated with TS. Cephalometric analyses have shown potential to improve the diagnosis and treatment of the occlusal and orthodontic/orthognathic disorders of young and adult subjects with TS [13,14,15], including those treated with GH. However, notable differences have been reported in the expected orthodontic results, either due to the TS oral ailments, such as reabsorptions and functional alterations, or to the different GH treatments applied [16,17].

As in many other rare and genetic diseases, a systemic and comprehensive description of the dental–oral–craniofacial phenotype in TS is lacking. The literature reports a notable number of cranial, oral and dental signs and symptoms. The different studies independently report abnormal craniofacial growth, a reduction in the transverse dimensions of the maxilla, mandibular sagittal deficiency, a high palatal arch, cleft palate, and malocclusion in women diagnosed with TS [18,19,20]. Tooth anomalies are also reported, including alterations in the chronology of tooth eruption and in tooth number, shape, size and alignment in the dental arch [21,22,23,24,25]. However, none of these reports analyze specifically, and simultaneously, all the possible oral phenotypical signs and symptoms of TS.

Considering the high number of phenotypic traits reported for TS and the possibility of individual or familiar aggregation, the purpose of our study was to elicit the dental, oral and craniofacial features that characterize the craniofacial phenotype in TS. A comprehensive analysis of the oral health status, as well as the craniofacial, oral and dental signs and symptoms of a cohort of TS subjects and first-degree relatives, was carried out to determine the more recognizable and distinctive features to be included in this phenotypical definition.

## 2. Materials and Methods

An exhaustive exploration of the dental–oral–craniofacial phenotype in TS was conducted, encompassing the analysis of girls and women diagnosed with TS and their first-degree relatives. The study was performed under the approval of the Institutional Review Board of the University of Barcelona’s Ethics Committee (IRB00003099). Subject records, anamnesis and clinical evaluations were obtained for all participants after signing the informed consent.

### 2.1. Study Sample

The analyzed cohort included 15 woman with TS aged from 3 to 48 years old. TS diagnosis was established by sex chromosome analysis. The sample also included 14 first-degree relatives (mothers and fathers of the subjects), who ranged in age from 32 to 78 years old. All participants were of European origin. Only the youngest TS case was in mixed dentition, and none of the study subjects showed evidence of periodontal disease.

The study design excluded individuals who had undergone maxillofacial surgeries that could alter the analysis of facial phenotype. Dental treatments were meticulously recorded and analyzed to ensure that they did not induce major changes in the observed phenotype. Although TS is a rare disease and the number of cases is scarce, we collaborated with the Catalan TS patient association, ASTCatalunya, to recruit as many individuals as possible. 

### 2.2. Dental Conditions Recorded 

The clinical data recorded for subjects with TS included the age at diagnostic, as well as other systemic signs and symptoms of Turner syndrome. General health anamnestic data were collected for all participants, including height, weight and posture alterations, as well as the pharmacological treatments and nutritional supplements received.

To assess the dento-craniofacial characteristics, we followed the Multifunction System (MFS) [26,27]. This system involves a qualitative and quantitative method of oral health exploration using oral disfunction anamnestic data, such as sleep disturbances (excessive sweating, salivation, respiratory difficulties or head posture alterations during sleep; snoring; sleep apnea and/or polysomnography records), as well as respiratory disturbance signs and symptoms (type of respiration, resting open-mouth posture or excessive salivation, allergies, asthma, frequent colds or voice loss, parafunctional oral habits, speech disturbances, exercise fatigue) [26,27]. 

Moreover, the participant’s oral health anamnestic report included the family and personal oral health history of all participants (dental treatments, orthodontics, oral surgery procedures, diet type), followed by an assessment of tooth anomalies in number, size, shape, structure, color, position, alignment or eruption pattern. The participant’s facial profile, lingual mobility, nose forced respiration pattern, tonsil size (or adenoidectomy/tonsillectomy history), labial dynamics, malocclusion (Angle Classification), crossbite/open bite and swallowing dynamics were also explored and recorded to provide a highly detailed description of the dental–oral–craniofacial phenotype associated with TS, both in the subjects and their first-degree relatives.

The anamnestic general health data relevant to oral health were grouped into two main blocks: data reporting sleep disturbances, and data about respiratory signs and symptoms. These data were specifically searched both on TS subjects and their first-degree relatives, in order to establish whether there was familiar aggregation, as described by Pinho et al. [28] and Tallón-Walton et al. [29]. Individual aggregation was calculated as the number of signs and symptoms present in each individual, which ranged between 0 and 13.

### 2.3. Statistical Analysis

Considering the limited sample size and statistical power of the study, we only performed exploratory and descriptive analyses of the oral/dental signs and symptoms observed in the TS subjects and their first-degree relatives. 

The means of all recorded data from the samples and their standard deviations were compared to identify phenotypical patterns among different subjects. 

## 3. Results

### 3.1. Clinical Findings

Respiratory signs and symptoms were reported by 42.85% of the subjects with TS, including tiredness related to physical exercise, open-mouth posture while watching TV or the PC screen, and speech disturbances. Additionally, 42.85% of TS subjects received speech therapy, while 71% revealed a history of complete or partial tonsillectomy. Frequent colds were reported by 28.57% of subjects, but no cases of respiratory allergies, asthma, voice loss, parafunctional habits, or excessive salivation were reported. Sleep disturbances included excessive salivation during sleep in 50% of the subjects, restless sleep (35%), snoring (33%), excessive sweating at night (21%), and sleep apnea (13%). None of the TS subjects reported experiencing abnormal postures during deep sleep. The oral functional exploration revealed a collapsed nostril configuration in 42.85% of respondents.

### 3.2. Dento-Craniofacial Findings

Regarding the craniofacial phenotype (Table 1), 85.71% of the subjects presented the TS characteristic convex facial profile (Class II facial profile), evidencing insufficient mandible development. Moreover, 42.85% of TS women were diagnosed as Class II division 1 malocclusion, with an increased overjet; meanwhile, 42.15% showed Class II division 2 malocclusion. Interestingly, most of the study participants’ family members (81.81%) showed a Class I normal facial profile, except one case showing a Class III concave facial profile (Figure 1).

Moreover, 35.71% of cases evidenced deep bite, 21.42% cross bite and 14.28% open bite. Dental crowding was evident in 57.14% of the TS subjects, even when all subjects reported dental treatments and regular follow up, including orthodontic treatment in 66% of the participants. Orthognathic surgery was reported during one subject’s exploration, who was thus excluded from the analysis. A total of 28.57% of the TS subjects presented more than 50% of the TS occlusal signs and symptoms when evaluated during the study. Three family members (21.42%) showed deep bite, but only one of them was related to a TS subject showing the same sign (Figure 2).

When explored, 71.42% of the subjects presented lip contact at rest, while 42.85% interposed the tongue or lips while swallowing. Moreover, 50% of TS subjects presented dysfunctional swallowing [26,27,30]. Only 42.85% of the participants were able to make contact between the tip of the tongue and the palate. One subject showed ankyloglossia, while one subject reported a frenectomy intervention. In total, 33.71% of the subjects showed signs of total or partial tonsillectomy. Finally, 57.14% of the TS subjects presented more than 50% of the TS oral signs and symptoms, while 21.42% presented the complete set of TS oral signs/symptoms described in the literature (100%). Again, none of the aforementioned signs and symptoms appeared in the relatives, who showed normal tongue mobility, and only in one case an atypical deglutition was evidenced.

Tooth anomalies of number, size, structure, shape, position and eruption [31] that are reported as associated with TS were present in most of the analyzed subjects: 57.14% presented less than 50% of the syndrome signs and symptoms, while 42.85% presented more than four out of the seven reported TS characteristic dental signs and symptoms (supernumerary molars, agenesis, microdontia, enamel defects, alterations in eruption patterns, including advanced and delayed for chronological age, crowding, rotations and transpositions) (Figure 3). Tooth number anomalies, including maxillary incisors agenesia and supernumerary molars, were found in 42.8% of the subjects. In total, 35.71% of the subjects displayed tooth size anomalies: two had upper incisors microdontia, two had generalized microdontia, and one subject presented molar microdontia. Enamel dysplasia, a tooth structure anomaly, was observed in 21.42% of the subjects and only affected the mandibular incisors. No tooth shape anomalies were observed. In addition to dental crowding, 35.7% of subjects reported eruption alterations and 21.42% presented premature eruption. 

Figure 4 compares the number and distribution of oral/dental signs and symptoms in the two groups of study participants, TS subjects and their first-degree relatives. In the whole cohort, we observed a minimum of 2 and a maximum of 10 signs and symptoms out of the 13 oral health signs reported in the literature as being characteristic of the syndrome. Moreover, we detected that in the group of TS subjects, the individual aggregation of oral/dental signs was larger than in the group of relatives. The mean number of oral health signs observed in the TS subjects was 6, and the most frequent number of signs was 10. In the relatives, the mean number of oral health signs observed was two, and the most frequent number of signs present in this group was five (Figure 4). 

The characteristic facial profile, as well as the tonsil alterations, the malocclusions and dental misalignments were found in most of the TS subjects, while the lingual mobility alterations, as well as the dental anomalies, were found in a restricted number of TS subjects. Of the aforementioned characteristic signs and symptoms of TS, the only one observed in 45.45% of the first-degree relatives was a small palatine tonsil size (less than 25% of the normal volume). Interestingly, most of the study participants’ family members (81.81%) showed an Angle Class I.

## 4. Discussion

The cranial, oral and dental phenotypic traits described in Turner syndrome subjects are numerous. The ORPHANET [32] list of clinical signs and symptoms under the Turner syndrome ORPHA code 881 reports at least seven signs and symptoms as “Frequent”, including a high and high narrow palate; micrognathia and retrognathia; and neck pterygia, a thickened nuchal skin fold and a webbed neck. Moreover, both “Abnormality of the dentition” and “Aplasia/Hypoplasia of the mandible” are mentioned as “Occasional” diagnostic findings. No other cranial, oral or dental signs are reported as “Very frequent” or “Rare”. The Human Phenotype Ontology database linked to ORPHANET provides, for each descriptor, the associated diseases and genes, as well as a short description of the phenotypical trait and a list of synonyms [33]. The synonyms describing each one of the ORPHANET morphological and physiological signs and symptoms do not reflect a coherent taxonomy [31]. Consequently, a systematic anamnesis by experienced oral health professionals in collaboration with a wide multidisciplinary healthcare team is required for an early and accurate confirmation of the TS signs and symptoms.

The MFS (Multifunctional System) Diagnostic Protocol described in 2003 by Duran and Ustrell [27] has become a multidisciplinary assessment resource for the examination of the oral, facial and dental characteristics of children, adolescents and adult subjects [30]. The systematic anamnesis and exploration of morpho-functional dental, oral and craniofacial alterations and the use of a common taxonomy [31] has enabled our team to report and assess the relevant signs of TS previously overlooked. These data can facilitate individual diagnosis and should be included in future multicentric studies [34] and databases that aim to develop advanced algorithm-based diagnostic systems [19,35].

Most of the congenital tooth anomalies associated with TS appear between 9 and 10 years of age, which makes this period critical to TS diagnosis. Moreover, tooth number anomalies, which in the general population have a prevalence ranging between 17% and 36%, without a significant difference between sexes [36,37,38], were observed in 42.8% of our TS cohort. Remarkably, none of the TS subjects’ first-degree relatives showed tooth number anomalies, in contrast with the familiar aggregation observed in tooth agenesia in general populations [28,39].

The incidence of microdontia in the general population is reported to be between 1.5 and 2% [36,37,38]. These values are notably lower than those observed in our Turner syndrome cohort, where the incidence was found to be 35.71%, with the upper lateral incisors being the most affected teeth. Again, only one family member of our TS subjects presented microdontia, in contrast with reports of familial aggregation [39,40] and genetic mutations leading to oral–dental phenotypical traits aggregation [41]. 

Various etiological factors are associated with damage to enamel development, resulting in enamel dysplasia. This term encompasses descriptors such as enamel hypoplasia, enamel agenesis, enamel hypo-mineralization, enamel hypo-maturation and amelogenesis imperfecta [31]. Manifesting primarily as diffuse enamel opacities, enamel dysplasia increases the risk of caries and tooth wear in both deciduous [42] and permanent dentition [43,44]. The prevalence of enamel dysplasia among children populations ranges between 6% and 55.2%, which matches our cohort’s prevalence (21.42%). In our cohort of TS subjects, enamel dysplasia was specifically located in the maxillary incisors, which are commonly reported as affected teeth in TS subjects [45]. Regarding the previously reported tooth shape anomalies in TS women [46], none were observed in our cohort.

The growth hormone has an important role in regulating the differentiation and development of dental organ tissues. Subjects receiving hormone replacement treatments have shown a higher incidence of tooth number and size anomalies [45]. Before therapy, a delay in the dental maturity and eruption pattern was described, along with a marked delay in bone development and a tendency towards dental crowding. A high risk of dental caries was described in this group of subjects, which decreased following an increase in Vitamin D levels [47,48]. The effects of hormonal treatment, particularly on the dental-oral–craniofacial region, strongly depend on factors such as the age at which treatment starts, its duration, the frequency of administration, and dosages [17]. This observation is consistent with the clinical history and phenotype of our cohort.

Recent retrospective studies have reported that between 24.1% and 24.4% of orthodontic subjects present one congenital tooth anomaly, while subjects presenting more than one anomaly range between 1.2% and 4.6% [49,50]. A cross-sectional study detected, in non-orthodontic subjects [51], an overall prevalence of tooth anomalies of 20.9%, of which 17.9% were unique; meanwhile, 2.7% reported two anomalies, and only 0.3% of subjects studied presented three or more anomalies. These data differ from the TS subjects analyzed in this study, who showed a minimum of 2 and a maximum of 10 of the 13 oral anomalies explored, with a median of six anomalies. Facial signs of TS were observed in 79% of the participants, while the expected tongue and tonsil anomalies were evident in only 21%. In our cohort, the overall prevalence of dental signs encompassed 43% of TS subjects presenting more than 50% of all the possible tooth anomalies, whereas 57% of subjects presented less than 50% of the analyzed tooth anomalies. Only 7% of TS subjects did not present any of the tooth anomalies. Microdontia, a common size anomaly reported in the TS cohort, was recently reported to be slightly more prevalent in women [52]. However, no other non-metrical dental anomalies associated with sexual dimorphism were found, either in our sample subjects or in their first-degree relatives. 

The concomitant presence of numerous dental, oral and facial phenotypic traits in the same young female was notable in our TS cohort. Remarkably, most dental phenotypical traits reported to exhibit familial aggregation in the general population, such as microdontia and dental agenesis [28,29,39,40], were absent in the family members of the TS subjects explored in our study. 

Despite the well-known clinical picture of the syndrome, this study is the first comprehensive exploration of the multiple oral health ailments affecting TS subjects. So far, no classification of the dental, oral and craniofacial anomalies of rare genetic syndromes has been found in the literature, despite the fact that most rare disease reports describe a high number of dental–oral and craniofacial phenotypical traits [53]. The scarce use of common taxonomy [31], systematic exploration protocols [27,30], clinical classifications [53] and oral health data [54,55] weakens the feasibility of providing oral health guidelines for TS subjects [56]. The active participation of Oral Health Specialists in the multidisciplinary health team must serve as the foundation for constructing comprehensive anamnestic procedures. These procedures are crucial to facilitating an early diagnosis and the successful clinical management of Turner syndrome.

## 5. Limitations of the Study

Our study presents several limitations that need to be carefully considered. First, the sample size analyzed is relatively small. This limitation may affect our ability to extrapolate our findings to a wider population. However, it is important to consider the context in which we are working, as we are investigating a minor condition that falls within the group of rare diseases. Considering the low prevalence of this syndrome, the size of our sample could be deemed considerable in comparison to other studies focusing on rare diseases. Nevertheless, it is important to acknowledge that a larger sample size would have provided greater robustness in our findings and would have allowed for a more comprehensive understanding of the phenotypic variability in Turner syndrome.

Second, it is relevant to mention that all participants in our study were of European origin. This fact could limit the generalization of our results to other populations. The ethnic homogeneity in our sample could bias the results and may not fully reflect the diversity of the population affected by Turner syndrome.

A third limitation to consider is the exclusion of individuals who have undergone maxillofacial surgery. This decision may have eliminated those with more severe phenotypes or those who could have had a significant impact on the study results. It is possible that these excluded individuals represent an important part of the phenotypic variability in Turner syndrome, and their absence could influence the representativeness of our results.

The literature descriptions of most of the dental, oral and craniofacial phenotypical traits associated with TS lack a coherent taxonomy, with a clear definition of the morphological anomaly reported. This limitation hinders the phenotypical analysis, making the early detection of TS and adequate referral challenging.

These limitations underscore the need for future research with larger and more diversified samples to gain a more complete and accurate understanding of this complex genetic condition.

## 6. Conclusions

The concomitant presence of numerous dental, oral and craniofacial signs and symptoms in one young girl must alert the Oral Health Team to identify this particularity of the TS phenotypical profile and refer the subject to the adequate expert center.

The absence of familiar aggregation of the dental, oral and craniofacial signs and symptoms must also alert the health team.

The timing of the congenital tooth anomalies reported makes the 9 to 10 years period critical for the diagnostic and clinical management of Turner syndrome in female subjects.

The clinical management of rare diseases must be carried out by multidisciplinary teams including a wide variety of Health specialists to ensure the comprehensive care of its numerous pathologies.

## Figures and Tables

**Figure 1 diagnostics-14-00769-f001:**
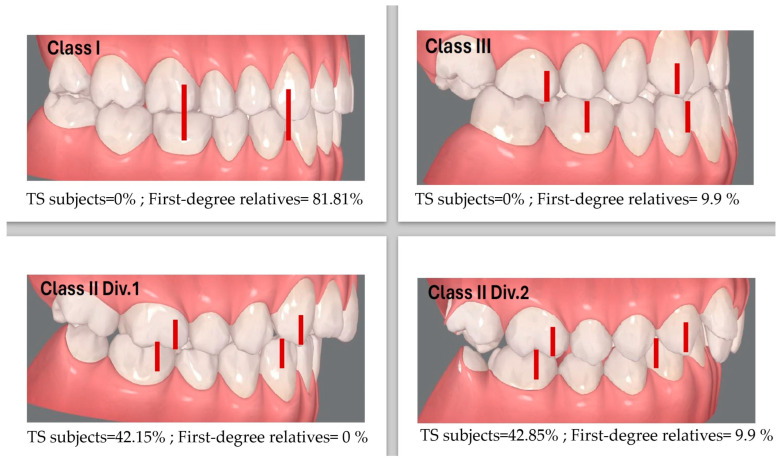
Prevalence of anteroposterior malocclusions in TS subjects and first-degree relatives.

**Figure 2 diagnostics-14-00769-f002:**
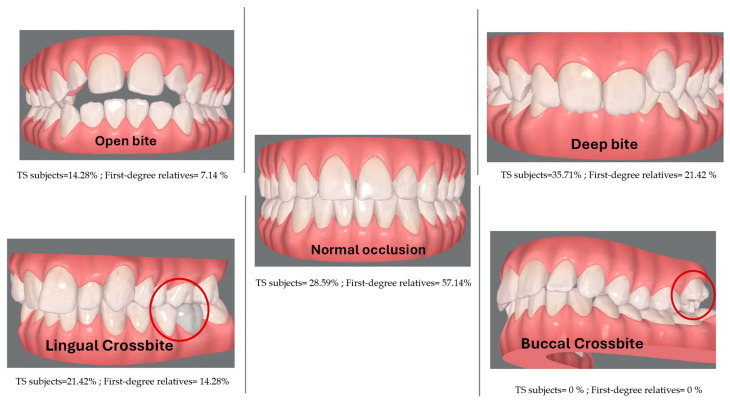
Prevalence of vertical malocclusions (**top row**) and transverse malocclusions (**bottom row**) in TS subjects and their first-degree relatives.

**Figure 3 diagnostics-14-00769-f003:**
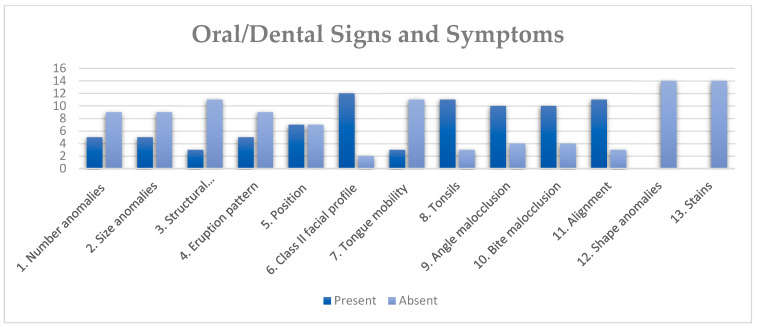
Number of TS subjects presenting or not presenting each of the 13 oral/dental signs and symptoms typically associated with TS, as described in De la Dure-Molla et al., 2019 [31].

**Figure 4 diagnostics-14-00769-f004:**
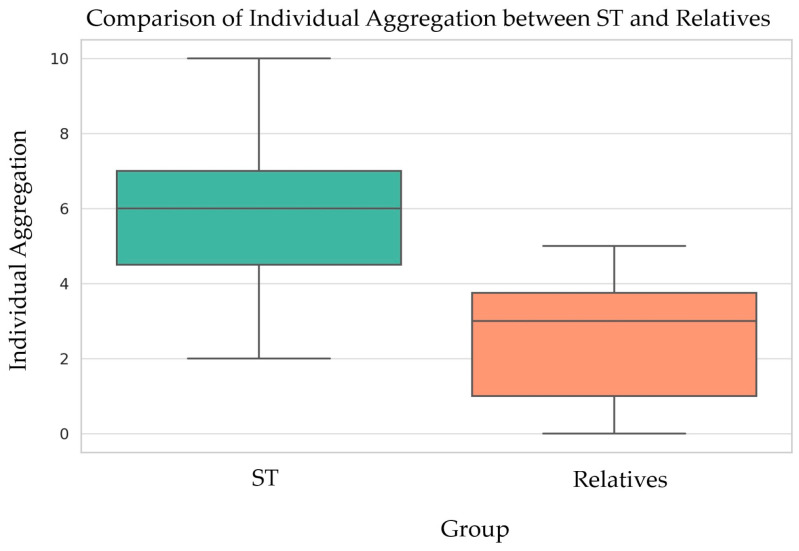
Comparison between the individual aggregation of oral/dental signs and symptoms assessed in TS subjects and their first-degree relatives.

**Table 1 diagnostics-14-00769-t001:** Relative percentage of appearance of Dento-Craniofacial signs and symptoms in each of the patients studied (+relatives).

ID	Number + Size + Shape + Structure + Position + Alignment			Malocclusion			Tongue Mobility + Tonsils				Class II Facial Profile		
TS Subject	% Signs	% First-Degree Relatives (Mother|Father)		% Signs		% in First-Degree Relatives (Mother|Father)	% Signs		% in First-Degree Relatives (Mother|Father)		% signs	% in First-Degree Relatives (Mother|Father)	
ESTM0003	50%	NA		100%		NA	50%		NA		0%	NA	
ESTM0016	33%	NA		50%		NA	0%		NA		100%	NA	
ESTM0021	0%	33%		0%		0%	50%		0%		0%	0%	
ESTM0022	50%	17%		17%	100%	50%	50%	50%	0%	0%	100%	0%	0%
ESTM0026	66%	NA		100%		NA	100%		NA		100%	NA	
ESTM0027	33%	NA		100%		NA	50%		NA		100%	NA	
ESTM0028	33%	50%		100%		100%	50%		0%		100%	0%	
ESTM0030	50%	NA		100%		NA	50%		NA		100%	NA	
ESTM0031	50%	0%	33%	100%		50%	0%	50%	0%	0%	100%	0%	100%
ESTM0034	33%	33%	17%	50%		0%	0%	0%	0%	50%	100%	0%	0%
ESTM0037	50%	17%		50%		100%	50%		0%		100%	100%	
ESTM0041	16%	17%	0%	0%		0%	0%	0%	0%	0%	100%	0%	0%
ESTM0045	33%	0%		50%		0%	100%		0%		100%	0%	
ESTM0047	16%	NA		100%		NA	100%		NA		100%	NA	
Average	37%	24%		71%		29%	50%		4%		86%	16%	

## Data Availability

Data collected on TS subjects are already provided in the article. Clinical images and data cannot be made available due to restrictions imposed by the ethics approval for personal data protection.

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
