# Peer review of "Comprehensive Oral Diagnosis and Management for Women with Turner Syndrome"

_diagnostics, 2024, doi:10.3390/diagnostics14070769_

Round 1
Reviewer 1 Report
Comments and Suggestions for Authors
Please see enclosed PDF for a point by point analysis

Author Response
Dear reviewers,
We would like to begin by expressing our sincerest gratitude for the time, effort, and dedication you have invested in reviewing our article. Your observations and suggestions have been invaluable in enriching and refining its content, ensuring that the article reaches its full potential.
We are pleased to inform you that we have accepted all proposed suggestions and have incorporated them into the text, faithfully following your directions, as you will see indicated in red. Each change has been implemented with the goal of clarifying, expanding, and improving the accuracy of the article, respecting the spirit and intention behind each of your recommendations.
Regarding the question about whether we have access to x-rays, or panoramic x-rays, no imaging tests are available since patient recruitment was conducted solely for dental examination purposes. It would be necessary to request a specific ethics committee to perform radiation tests on the study subjects. We believe this could be very interesting for future studies.

Reviewer 2 Report
Comments and Suggestions for Authors
Paper needs revision by a native English speaking as there are many typos and grammar issues as well as vague terminology
Abstract, too much background, remove 'Recently, several studies highlighted the relevance of oral health in TS and have provided scattered 24
evidence of oral, dental and craniofacial anomalies associated with TS, yet a comprehensive description is still lacking.'
add the age range, 3 to 48 year
you mentioned a wide age range of 3 to 48 year, you need to add how many of them were in mixed dentition and how many in adult dentition and then add the details of Tooth anomalies (what are they?, maxillary incisors agenesia and supernumerary molars were found in 42.8%, microdontia ), malocclusion (what types), and Class II facial profile, while relatives exhibited fewer manifestations.
Literature review, focus on TS only, remove other conditions
Methodology, add subheadings for the study sample, variables/conditions recorded (dental/medical), statistical analysis
did you have access to x-rays, or panoramic xrays?
Results
add separate subheadings for medical and dental findings
line 142, this needs revision, what is the 3A classification, add reference ' This symptomatology is consistent with the oral functional exploration, which revealed a Grade 3A (bilateral) collapsed nostril configuration in 42.85% of the respondents, while the rest of the cohort exhibited an unilateral nostril collapse (Grade 2) or immobility of the nostrils (Grade 1) during forced breathing. '
line 285, remove 'Tooth shape anomalies were neither reported nor observed during the exploration. ' as you mentioned microdonia before
Figure 3, add footnotes for 13 elements, expand each and describe what you mean, in particular, for numbers 1, 4,5,7(tongue tie?), 9 (what do you mean?), 10(again what do you mean?)
Figure 4, needs revision, I could not grasp what it tries to show, enlarge and add more information
Discussion
1st paragraph needs citations
line 241, 'upper lateral incisive ' did you mean 'upper lateral incisor(s)'?
line 262-264, add the following as well in orthodontic patients (Acta Odontol Scand. 2011 Mar;69(2):125-8.;J Oral Sci. 2010 Sep;52(3):455-61.) and compare your findings with them
needs major revision
Author Response

(The authors gave the same response as above.)

Round 2
Reviewer 1 Report
Comments and Suggestions for Authors
The manuscript has been improved
Author Response
Dear reviewer:
Thank you once again for your valuable contribution to the refinement of our work. We are eager to advance this manuscript to the next stages of publication and believe that your insights have greatly contributed to its improvement.
Warmest regards
Reviewer 2 Report
Comments and Suggestions for Authors
Thank you for the revision: you need to revise the paper for presentation an style, some examples are followed
line 99, add female /male number
line 115, change to 'Dento-craniofacial characteristics'
line 125, correct to 'followed by assessment of tooth anomalies in number, size, shape, structure, colour, position, alignment or eruption pattern'
line 128, correct to'Angle Classification'
line 141, remove 'However, considering that TS is a rare disease, 141
our preliminary results are relevant for future studies. '
line 150, correct to 'Clinical findings'
line 164, correct to 'Dento-craniofacial findings'
line 167, correct to 'Moreover, 42.85% of TS women were diagnosed as Class II division 1 malocclusion, with an increased overjet; whereas 42.15% showed Class II division 2 malocclusion. Interestingly, most of the study participants' family members (81.81%) showed a Class I normal facial profile, except one case showing a Class III concave facial profile'
line 176, correct to ' had deep bite'
line 182, correct to' showed deep bite'
Comments on the Quality of English Language
needs revision
Author Response
Dear Reviewer 2,
I hope this letter finds you well. I am writing to express our heartfelt gratitude for the time and effort you dedicated to reviewing our manuscript. Your insightful comments and constructive suggestions have been invaluable in enhancing the quality and clarity of our paper.
I am pleased to inform you that we have carefully considered and implemented all the changes you suggested. Allow me to summarize the adjustments we made in response to your recommendations:
- Presentation and Style Revision: We have thoroughly revised the paper for presentation and style as advised.
- Line 99: We have added the number of female and male participants as suggested.
- Line 115: The term has been updated to 'Dento-craniofacial characteristics'.
- Line 125: The description has been corrected to 'followed by assessment of tooth anomalies in number, size, shape, structure, color, position, alignment, or eruption pattern'.
- Line 128: Corrected to 'Angle Classification'.
- Line 141: The suggested text has been removed to streamline the focus on the relevance of our preliminary results for future studies.
- Line 150: Updated to 'Clinical findings'.
- Line 164: Now reads 'Dento-craniofacial findings'.
- Line 167: Adjusted to accurately reflect the findings as 'Moreover, 42.85% of TS women were diagnosed as Class II division 1 malocclusion, with an increased overjet; whereas 42.15% showed Class II division 2 malocclusion. Interestingly, most of the study participants' family members (81.81%) showed a Class I normal facial profile, except one case showing a Class III concave facial profile'.
- Line 176 and 182: Both lines have been corrected to 'had deep bite' and 'showed deep bite', respectively, to maintain consistency in the terminology used throughout the manuscript.
We believe that these amendments significantly improve our manuscript, not only in terms of precision and readability but also in the clarity of the findings presented. Your meticulous review and constructive feedback have been instrumental in achieving this enhancement, and for that, we are profoundly thankful.
Please rest assured that we have taken great care to ensure that all your suggestions have been fully addressed. We are hopeful that these revisions meet your approval and look forward to any further guidance you may offer.
Thank you once again for your valuable contribution to the refinement of our work. We are eager to advance this manuscript to the next stages of publication and believe that your insights have greatly contributed to its improvement.
Warmest regards
Round 3
Reviewer 2 Report
Comments and Suggestions for Authors
Thank you for the revisions
Comments on the Quality of English Languagepaper needs professional editing by a native English speaking clinician